# Implications of Renovated Buildings in Yeonnam-Dong, Seoul, an Area under Commercial Gentrification

**Dai Whan An** [1] and **Jae-Young Lee** [2,*]

1   Department of Architecture, Chungbuk National University, ChungDaeRo 1, Seowon Gu, Cheongju-si 28644, Republic of Korea
2   School of Architecture, Hongik University, Seoul 04066, Republic of Korea
*   Correspondence: ljy21@hongik.ac.kr

**Abstract:** We aimed to identify the characteristics of the changes in the buildings and alleyways in Yeonnam-dong, Seoul, where low-rise, residential buildings are being renovated or repurposed into commercial buildings, as well as to investigate their renovations, repurposes, and sociocultural implications. Thus, we surveyed and classified 149 renovated buildings, investigated the perceptions and ambiance of the area and buildings using a trade area analysis and interviews with visitors and store owners, and uncovered the importance of renovation. Since the early gentrification, a trend of performing renovations that retained the original form of the building from the initial renovation stage was seen; this created an ambiance of nostalgia, naturalness, and authenticity, along with the urban conditions of low-rise, residential buildings in Yeonnam-dong, a representative undeveloped area. These renovated buildings reflect the social status, taste, and practice of gentrifiers, and they reveal a hybridization of the past and present, Korean circumstances and exotic cultures, and residential and commercial buildings. As commercialization progressed, renovated buildings vastly differing from the original and displaying active commercial characteristics were seen. Our findings imply that the area's early ambiance, which had an air of "distinctiveness", has lost its personality and begun to generalize. Thus, numerous gentrifiers have been replaced and several aspects of renovation have changed that the visitors are aware of.

**Keywords:** commercial gentrification; renovation; authenticity; commerciality; ambiance

## 1. Introduction

### 1.1. Background and Purpose

The commercial district of Yeonnam-dong was once a residential area dominated by lower-middle-class housing. It is located in the northwestern part of Seoul and was planned as a residential area. It neighbors the Hongdae area (Hongik University), which has been rapidly commercialized since the 1990s. As the Hongdae commercial area began to expand in 2008 [1], Yeonnam-dong was commercialized, and, since 2014–2015, with the completion of the Gyeongui–Jungang Line and the Gyeongui Line Forest Park, the commercialization of residential areas has increased rapidly. Old buildings and meandering alleys in the areas that were excluded from urban development were rediscovered. With shops, such as cafés, restaurants, clothing boutiques, and jewelry stores, arrayed along the alleyways, it soon became a prime location and new center of cultural consumption for young people. Thus, the existing residential buildings were repurposed and renovated as commercial buildings leading to commercial gentrification. Hence, this side-street trade area that now has intermingled commercial and residential uses has come to represent a unique place. This urban phenomenon revitalizes underdeveloped areas as spaces of cultural consumption, and the increase in the number of customers leads to a rise in the rent, thereby continuing commercial gentrification, which drives the initial gentrifiers to look for other outskirts in search of lower rent; this phenomenon is called displacement (Figures 1 and 2) [2].

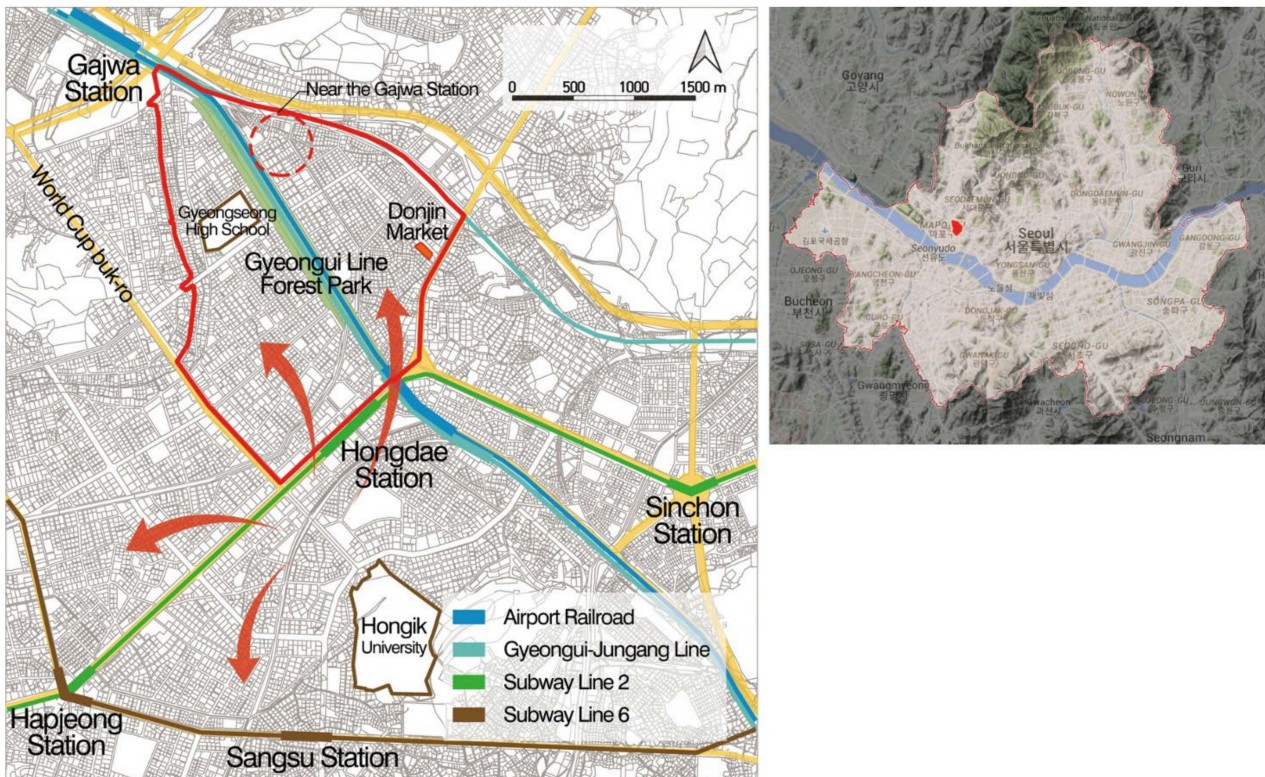

**Figure 1.** Study area (area bordered in red) and gradual commercial expansion around the Hongik University.

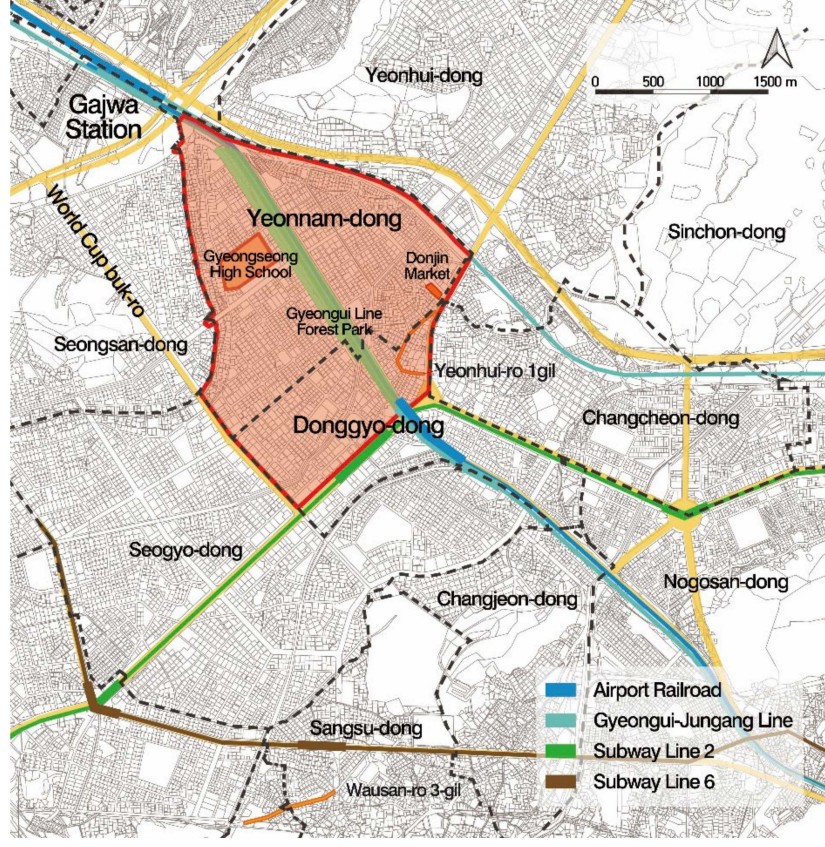

**Figure 2.** Administrative districts in and around the study area.

Many sociological and urban studies have been conducted on the phenomenon of commercial gentrification in this area. The main focus of urban studies has been on the quantitative analysis of cities and their management systems at the urban scale, along with the sociological studies on the tastes of gentrifiers and displacement. Few studies have been dedicated to exploring the changes in buildings and alleyways due to gentrification and analyzing their implications. It is worth regarding the buildings themselves in the area for interpreting the ambiance or placeness. The local ambiance is the result of the interactions between the activities of the shop owners and visitors and transformation of buildings and alleyways. This study aims to answer the following questions: How have the buildings and streets changed? What charms the visitors about the transformed buildings and streets? How do store owners perceive the area and renovated buildings? In the stage of commercial gentrification, how is the local ambiance perceived between authenticity and commerciality? What are the implications of renovated buildings in the sociocultural, urban, and architectural aspects? To this end, we analyzed the repurpose and renovation of buildings in the Yeonnam-dong area and examined their implications, which generate the ambiance and placeness.

### 1.2. Theoretical Review: Gentrification, Authenticity, and Commerciality

Ruth Glass first coined the term "gentrification" in reference to the influx of the middle class into working-class residential areas, upgrading of their houses, and displacement of residents [3]. Hence, she observed the involvement of movement of classes in urban change. Subsequently, Neil Smith explained the cause of this urban phenomenon according to the theory of capitalist society's uneven development [4]. He also explained the forms of gentrification as three "waves" from a historical perspective: urban revitalization policies led by the state, increased investments in urban residential districts led by the market, and aggressive intervention by large capital [5]. This can be viewed as a theoretical response to reflect these various gentrification phenomena.

Among the various concepts and discussions on gentrification, it is necessary to consider regional specificity. Discussions based on gentrification studies conducted in Europe and the United States so far may not be applicable in Asian contexts. For example, David Ley and Sin Yih Teo's work on the concept of gentrification in Hong Kong shows that there are not many discussions that view gentrification negatively, taking into account displacement and housing dispossession, as in Europe or the United States. In fact, urban redevelopment undertaken by developers is frequently perceived as neutral, if not positively, with an image of compensation and the good life. This is seen as evidence of the existence of alternative Asian modernities in urban development, operated by the state and the market [6]. As such, regional specificities can make a difference in the general concept and progress of gentrification. In Korea, residential gentrification through state-led apartment construction projects began to be studied since mid-2010s. These studies have contributed to the understanding of the developmental state and the role of real estate speculation in gentrification [7]. They showcased a two-sided perception of East Asian gentrification cases. Commercial gentrification, the subject of this paper, comprises different subjects from state- and market-led residential gentrification; however, commercial gentrification also shows a two-sided perception of gentrification. The primary reason is the difference in the position of members of society between the positive aspects of urban regeneration and development, and the negative aspects of the loss of authenticity and placeness amid rising rents and land prices due to commercial gentrification [8].

Commercial gentrification is when commercial facilities in commercial streets and areas become more advanced and "boutiquing". Small neighborhood stores in the area are replaced by galleries, cafés, luxury restaurants, and clothing stores, creating a unique placeness and boosting commercial activities [9,10]. Accordingly, large-scale capital flows in as rents rise, and the placeness of the region alters as stores become franchised and homogeneous. This phenomenon is called commercial gentrification [10]. Commercial gentrification in Korea mostly occurs in residential areas, not commercial areas, and, at

times, in semi-industrial areas (Seongsu-dong). Furthermore, it is difficult to explain the gentrification that has occurred since the mid-2000s in Yeonnam-dong, as well as Seongsu-dong, Gyeongnidan-gil, Samcheong-dong-gil, and Garosu-gil, in Seoul, through state-led or market-led forces or large capital intervention in terms of agent or scale. Considering the agents, it is more realistic to explain this gentrification in terms of the new middle class [11], the creative class based on the concept of the "creative class" of the characteristics of Yeonnam-dong area according to its activities and perceptions by the study of Park E.S. [1], and new small business owners, even if the characterization of these agents is ambiguous. In South Korea, there is a clear difference between traditional small business owners and new small business owners. New small business owners pursue exotic and hybrid store concepts with high cultural capital (educational background, career, overseas experience, etc.) [12]. In Yeonnam-dong's case in particular, artists, cultural planners, craftspeople, baristas, chefs, and the like were the early gentrifiers [1]. They created new shops, renovated the exteriors of buildings, and introduced a certain "landscape" in the area. This is referred to as a "gentrification landscape" and "gentrification aesthetics" [2].

These physical changes in the area express the identity of this new middle class. In Melbourne, residential buildings converted from Victorian, historical buildings were introduced [13] and, in Soho, New York, factory lofts were renovated to reflect the tastes of the artist community [14]. As the collective practices of gentrifiers, these actions reveal their identity by socially and spatially distinguishing them from other social groups [2]. According to Sharon Zukin, consumption activities in cafés, restaurants, and boutiques that reflect alternative cultural tastes of artists and consumers, who have aesthetic senses and refuse mainstream consumption culture during the gentrification process, are called consuming authenticity. This shows their differentiated tastes and aesthetics [15]. Through this process of gentrification, the economic and social position and cultural tastes of gentrifiers are realized in the physical landscape; in other words, the realized landscape carries social and cultural implications. Moreover, the agents based in this area, their activities, the influx of outsiders, and changes in the landscape create a sense and form the area's placeness or ambiance. Zukin states that, as shops pursuing economic opportunities enter through a new commercial area, new "images" and places in the area are formed [10]. This is because "places" are entities where objects and characteristics in the world are experienced within their implications [16].

However, as gentrification progresses gradually, people question the area's "authenticity", i.e., they voice the opinion that "this is not the area's true image". Authenticity is formed by the experiences of local members, and a place's identity is projected into everyday spaces. Zukin stated that authenticity also involves the history of a specific area or product, through which the consumption spaces fabricate an "aura of authenticity" [15]. Just as the first residents feel changes in the area's identity as early commercial gentrification causes residential areas to change, as commercialization progresses, new and stylish shops are established, the early gentrifiers are displaced, the area transforms once again, and the early gentrifiers question the authenticity of the area they had pursued. In other words, the act of questioning authenticity signifies changes in the area's commercial activities, physical changes in the area, and changes in experiences of the area due to agents.

The issue of authenticity in commercial gentrification is deeply related to commerciality. As commercialization progresses, authenticity is questioned. However, it is important to remember that the early gentrifier's activities in the area were also commercial. The characteristics of environments and activities in the area change sway between authenticity and commerciality while becoming a mixture of the two. Commerciality is also a characteristic of the area, a factor forming the gentrification landscape, and the "expression" of gentrification. The renovated exteriors, signs, and decorations placed in outdoor spaces are meant to attract passersby with "commercial charm" and reveal the place's character in its concrete form. Just as Robert Venturi analyzed the expressions and symbols of commercial buildings and signs in Las Vegas and assessed the value of architectural communication [17], the gentrification landscape is also an aspect that embodies the "everyday" activities of the area

through commercial vernacular, carries their significance, and reveals placeness. Therefore, authenticity and commerciality mixed in one area, along with urban and architectural objects, are experienced in their political, economic, social, and cultural sense.

Several examples of gentrification worldwide that created a place using the aesthetics of authenticity are available. Guillaume Sirois showed how the aesthetics of gentrification shown in the boutiques of Montreal's Mile End District created the value of authenticity. Regionally designed products mean a break in the global market economy for buyers, but he claimed that they further promote the momentum of neoliberal globalization. Sirois called it "selling authenticity" [18]. As such, authenticity and commerciality share a sensitive relationship. The desire for authenticity generates the aesthetics of anti-gentrification, but the aesthetics of anti-gentrification may operate an exclusive posture [19] or, as Sirois claimed, may not recognize that the desires and actions are part of a neoliberal world economy [18]. Therefore, it is necessary to examine in detail the ambivalent contradictions surrounding authenticity from various aspects.

### 1.3. Subjects and Methods of Research

Our study area includes Yeonnam-dong and some parts of Donggyo-dong. For covering the commercial areas in Yeonnam-dong, the study area was set as the area enclosed by the roads extending from the area between the northern Gyeongui–Jungang Line to the vicinity of the Gajwa Station to the north, World Cup buk-ro to the west, and Yanghwaro to the south. Administratively, the study area covers Yeonnam-dong and a part of Donggyo-dong (Figures 1 and 2).

In the building registration data of the study area, for the purpose of this study, we selected the buildings recorded as repurposed and renovated from 2000 onward. The first survey data were collected by the authors and five students from December 2018 to April 2019. The original street-side commercial buildings and buildings repurposed as guest houses (n = 68) were also excluded from our analysis. Although guest houses are also important buildings indicative of the commercialization of the area, they were excluded from the analysis because, in most cases, they were existing houses that were repurposed as lodging facilities (without renovation) and did not provide any valuable data with respect to the changes made to the buildings. We collected the data necessary for the study, such as the construction year, their previous purposes, repurposed use, number of floors, renovation year, store type on each floor, and post-renovation architectural features, of the 149 buildings surveyed, from the cadastral ledgers of the study area and through a field survey. As this study only considered recorded cases of renovations exceeding major repairs, we excluded cases where stores were opened after small-scale renovations and due to change of use. Therefore, there are more cases of commercial gentrification in the study area. Nevertheless, 149 cases were judged to be sufficient for the purpose of this study. The pre-renovation state of each building was extracted from the road view of the Daum portal site [20]. The 149 buildings were then divided into four categories according to the year of renovation, thus completing the map. This step was performed to confirm the locations of the surveyed buildings, identify the characteristics of the buildings and the surrounding area (shapes of streets and lots, relationships with the surrounding facilities, etc.), and determine the process of spread of commercial gentrification and current commercialization conditions. The second survey was a trade area analysis, in which the authors and two students investigated the commercial activities in the surveyed buildings via the internet and additional field surveys in August and September 2019. In the third and fourth surveys, the authors and three students interviewed visitors and store owners in January and February 2020 and December 2021. The surveys were carried out in the form of short interviews to qualitatively understand perceptions of the Yeonnam-dong area and changes in the landscape.

The survey subjects consisted of 31 visitors and five store owners. In total, 19 of the visitors who responded to the survey were in their 20s, along with 11 in their 30s and one in their 60s; there were 18 males and 13 females. The main questions asked were

the following: "What makes you come to Yeonnam-dong?" "What are the characteristics of Yeonnam-dong?" "How do you feel about Yeonnam-dong's alleyways and renovated residential buildings?" The stores for the survey were selected from within three categories of the surveyed buildings: one store in home/shop combination, two stores in buildings partially renovated with the original form retained among single-family housing, and two stores in buildings that underwent large-scale renovation with the original form partially retained among multi-family housing. The main questions were the following: "When will the store open?" "Why did you open a store in Yeonnam-dong?" "What is the ownership structure of the building?" "What do you think about Yeonnam-dong's ambiance?" "What are your intentions for the building's exterior renovation and interior?"

The 149 buildings surveyed were constructed from 1966 to 2011. They were mostly constructed in two periods: 1970–1974 (n = 45) and 1989–1992 (n = 48). The first case of renovation was conducted in 2004; however, most of these buildings were renovated between 2014 and 2018: five buildings in 2011, three in 2012, five in 2013, 11 in 2014, 22 in 2015, 18 in 2016, 40 in 2017, and 25 in 2018, indicating that intensive and rapid commercial gentrification took place in this area between 2015 and 2018 (Figure 3).

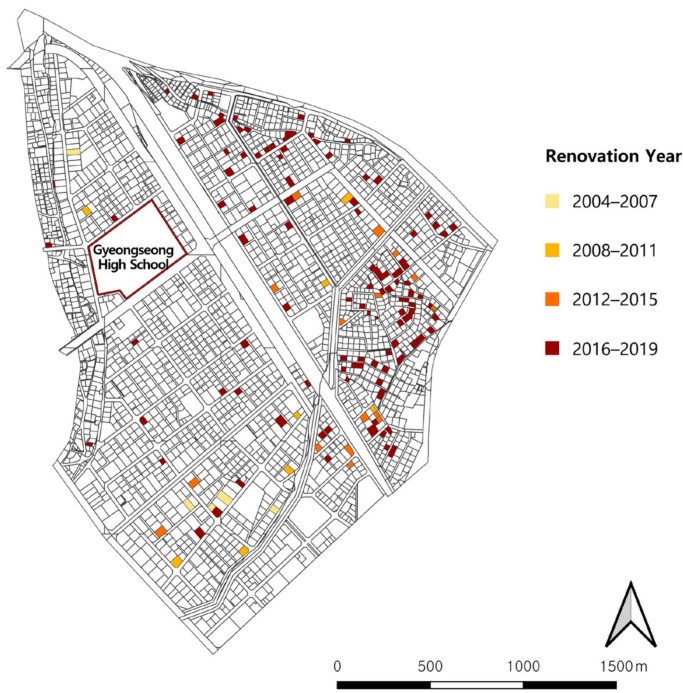

**Figure 3.** Distribution of renovated buildings by year.

For the purpose of this study, i.e., to identify commercial activities in the area, we investigated characteristics of the commercial activities of the buildings surveyed on the basis of the classification of the store types. Through this analysis, we could identify the degree of commercialization or the stage of commercial gentrification in Yeonnam-dong. Furthermore, we examined the changes in the exterior appearances and interior spaces of the renovated buildings and alleyways by classifying the renovation types. For the classification of the renovated buildings, the original buildings were categorized into three types according to form and then further classified into three groups according to the scale of renovation. This is because the original building's form and characteristics affect the nature of renovations and scale of investment. We then summarize the content of the interviews with the visitors and store owners and interpret them in connection with changes in the area and surveyed buildings. For more in-depth research, it is necessary to select the survey subjects in a more organized manner according to the degree of building renovation and nature of the industry, as well as increase the number of subjects. However, the analysis

of 149 buildings and surveys of 31 visitors and five store owners were judged to be sufficient to identify the ambiance of the area and buildings and their importance. Through this interpretation, we identify the relationship between commerciality and authenticity in the process of commercial gentrification and the ambiance and placeness of Yeonnam-dong, as well as uncover its social and cultural implications.

## 2. Formation and Gentrification of the Yeonnam-Dong Area

### 2.1. Formation of the Yeonnam-Dong Area

The current cadastral features (such as lots and roads) of Yeonnam-dong and Donggyo-dong were established within the framework of the Seoul Land Readjustment Project carried out during the late 1960s and early 1970s (1967–1971 Yeonhee Land Readjustment Project and Seongsan Land Readjustment Project). The southern part of Yeonhui-dong was separated in 1975 as an independent neighborhood (Yeonnam-dong means a neighborhood south of Yeonhui-dong). After the Land Readjustment Project, Yeonnam-dong and Donggyo-dong established themselves as typical residential areas in Seoul. They are still registered as type 2 general residential zones. A grid consisting of blocks was readjusted between the contexts organized by the irregular and narrow roads in existing urban areas. The new grid consisted of 6–20 m wide roads and parcels where one- or two-story single-family houses could be constructed. Therefore, the houses built up to 1984 have a typical layout, with a courtyard within the fenced parcel. Since 1985, however, with the application of relaxed regulations for multifamily houses, multifamily buildings having no courtyard for the maximum possible floor area ratio were constructed predominantly. Both multifamily housing and multihousehold housing have the characteristics of apartment houses in which several households live in one building. Legally, however, the former is a detached house with a single ownership and the latter an apartment house with multiple ownerships. In terms of size, a multifamily house has a floor area up to 660 $m^2$ and three levels, whereas a multihousehold house has a floor area of 660 $m^2$ or larger and can cover up to four levels. Despite these legal differences, they were classified as the same type for the purpose of this study because they have a layout distinct from a single-family detached house having a sufficient courtyard space. Moreover, most of the buildings surveyed were multifamily buildings. Consequently, the dominant building types of this area are two- or three-story, single-family and low-rise, multifamily houses.

Yeonnam-dong and Donggyo-dong are located on the outskirts of the expanding cultural consumption zone centered around the Hongdae area. Yeonnam-dong, which is now known as a side-street trade area, expanded from the Hongdae area and is a middle-class residential area well-known for many restaurants for taxi drivers. The Hongdae area itself was a residential area established in the 1970s. The area around Hongik University, a renowned fine arts college, began to become a cultural center, attracting ateliers, galleries, and small theaters. In the 1990s, the phenomenon of urban spatial culture began to take shape through the emergence of underground club culture, including live music theaters and dance clubs. Since the end of the 1990s, the Hongdae area has evolved into a major center of diverse cultural activities and entertainment, going beyond the scope of art and music, with an increasing number of new buildings constructed to accommodate different professionals in the cultural industry, including design, advertisement, film, broadcasting, photography, publishing, comics, fashion, and virtual entertainment. Young artists, bohemian intellectuals, and hipster-style cultural consumers gathered, creating a mecca of so-called indie culture, and, under their influence, this once quiet residential area has been transformed to a haven of alternative culture offering small and unique restaurants, cafés, bars, live music theatres, and dance clubs. However, exposed to the fierce process of gentrification since the mid-2000s, the Hongdae area has been filled with franchise coffee shops, fashion brand stores, and large night clubs. Moreover, driven by the recent increase in foreign tourists, duty-free shops, guesthouses, and discount cosmetics stores have opened in this area as well, adding to the characteristics of an entertainment area, thereby diluting its image as a cultural area. As the process of commercialization intensified

rapidly, the initial gentrifiers who made the Hongdae area a cultural center could no longer afford the soaring rent and moved to the surrounding area, which was then commercialized, pushing the gentrifiers further to the surrounding area, where gentrification continued. In this manner, the Hongdae area has been gradually expanding to the surrounding areas. In particular, the development of the Sangsu Station (on Line 6 in 2000), Hongik University Station (on the Airport Express Line in 2010), and Hongik University Station (on the Gyeongui–Jungang Line in 2014) accelerated the expansion of the Hongdae area by increasing the influx of the floating population.

Commercial gentrification in Yeonnam-dong began in 2008 from the south and southeast areas of Gyeongseong High School due to the Hongdae commercial area's expansion. From around 2012, young entrepreneurs and artists who could not afford the high rent in the Hongdae area began to flock around Dongjin Market in Yeonnam-dong, because the exodus of Hongdae nomads to Yeonnam-dong began around 2012–2013 [21]. The opening of the Hongik University Station on the Airport Express Line (2010) and Gyeongui–Jungang Line (2012–2014) and completion of the Gyeongui Line Forest Park (2015) led to a rapid commercialization of the Yeonnam-dong area, resulting in a rise in the rent (some of the initial gentrifiers left the area in search of affordable rent). The commercialization of the areas between Dongjin Market and the Hongik University Station on Gyeongui–Jungang Line has been even more intensified. According to the 2013 survey in the study by Park E.S. (Park E.S. 2013), there were only a few stores in the block between Dongjin Market and Hongik University Station on the Gyeongui–Jungang Line. However, the 2018–2019 survey study indicated that there are many renovated buildings in this area, suggesting that rapid commercial gentrification has occurred in this area since 2015 [1]. This commercialization is currently expanding to near the Gajwa Station (northeast end of the study area), and, as of 2020, the commercial gentrification of the area has expanded to the northwestern border of Yeonnam-dong (near the Gajwa Station), making the gentrifiers seek the most affordable rent. Thus, the initial gentrifiers are being pushed away from the core commercial area, and there is also a joint movement against displacement [22]. The area is home to a chain of gentrification, along with the displacement of merchants and craftsmen to the outskirts of Yeonnam-dong. Spurred by the expansion of the Hongdae area, the Yeonnam-dong residential zone has been transformed into a new cultural consumption venue, creating a side-street trade area through the repurposing and renovation of old single-family houses and multifamily buildings into shops and changing the street landscape, with fences demolished to provide access to shops.

### 2.2. Commercial Activities and Gentrification in the Yeonnam-Dong Area

Yeonnam-dong was a middle-class, residential area comprising low-rise, residential buildings. There were commercial properties in the area, such as Chinese restaurants, low-priced restaurants for taxi driver, and convenience facilities. Commercial gentrification began here in 2008, when shops began to move to Yeonnam-dong because of the escalating rent in the area in front of Hongdae. Hence, as the Hongdae commercial district expanded, the commercialization of Yeonnam-dong situated outside this area started. In Yeonnam-dong, the sides of the main roads outside the Dongjin Market block in the areas south and southeast of Gyeongseong High School were commercialized. Artists, musicians, chefs, designers, publishers, and media professionals who were active in the area in front of Hongdae settled in Yeonnam-dong in search of low rent. Yeonnam-dong was an area with ties to various regions of Mapo-gu with numerous publishers, Yeouido and its mass media companies, and the publishing complexes of Paju and Sangam. Since rapid commercialization began in 2015, many related offices have left Yeonnam-dong after the increase in rent.

In the initial period of gentrification, changes in streets and lots within Yeonnam-dong were not very prominent [1]. According to the distribution of 224 creative workforce-related industries and small-capital stores surveyed by Park E. S. in 2013, there were cafés (58), restaurants (16), pubs (13), small-cap stores and offices (10), ateliers and studios (16),

workshops (16), design offices (19), publishing houses (60), and community and cultural complexes (eight). Thus, the offices, workshops, cafés, and restaurants were found to coexist.

The research of Yoon Y. C. and Park J. A. provides more information on the degree of commercialization using the surveys conducted in 2014. Regarding the commercial gentrification stages, rather than Zukin's four-stage process of commercial gentrification where the area transitions from the final stage to luxury residencies, the three-stage classification of Yoon Y. C. and Park J. N. that analyzed South Korean data was judged to be more suitable [23,24].

They measured the degree of commercialization of 15 streets where commercial gentrification occurred in Seoul. Use of building, residential density, chain store ratio, and diversity of business type were combined to classify the degree of commercialization into Stages 1–3. Of the 16 streets, Yeonhui-ro 1-gil is within the area of this study. Although not a subject of this study, Wausan-ro 3-gil of Sangsu-dong, an area in the Hongdae commercial district where commercial gentrification occurred earlier than in Yeonnam-dong, is compared with Yeonhui-ro 1-gil. According to the study, Yeonhui-ro 1-gil and Wausan-ro 3-gil had the highest residential ratios and almost no chain stores. Although Yeonhui-ro 1-gil has nonalcoholic beverage cafés and Western-style restaurants, it also has a high proportion of Korean-style restaurants, hair salons, and hardware and heating equipment retailers, and was, thus, evaluated as Stage 1. Wausan-ro 3-gil shows a relatively high proportion of nonalcoholic beverage cafés and Western-style restaurants and also has retail clothing stores; thus, owing to its higher diversity of business types, it was evaluated as Stage 2. Hence, according to their study, the Yeonnam-dong area was at the initial stage of commercial gentrification in 2014. However, by 2018–2019, when the survey of this study was conducted, Yeonhui-ro 1-gil had undergone rapid commercial gentrification since 2014–2015 and greatly differed from that in 2014. According to the survey of this study, many buildings on this street were renovated, Western-style restaurants accounted for a high proportion, and the number of clothing and cosmetics stores increased. As of 2019, however, the commercialization of Yeonnam-dong has not yet reached Stage 3, in which clothing, cosmetics, and chain stores account for a high proportion of the total number of stores and the diversity of business types rapidly decreases (Tables 1 and 2). According to the study of Yoon Y. C. and Park J. A., among 16 streets in Seoul, only Garosu-gil was evaluated as Stage 3 [23]. Only by continuing to observe the Yeonnam-dong area will we know whether it progresses to Stage 3, e.g., Garosu-gil in the research of Yoon Y. C. and Park J. A. This is because Yeonnam-dong and Garosu-gil have different regional characteristics.

**Table 1.** Trade area analysis.

| Sector | Food | Retail | Living | Health | Education | Leisure | Sum |
|---|---|---|---|---|---|---|---|
| No. of stores | 820 | 237 | 134 | 41 | 30 | 19 | 1281 |

Trade area analysis of Yeonnam-dong (in the part circuited by the author), KB Real Estate Liiv ON https://onland.kbstar.com/quics?page=okbland (accessed on 18 November 2020).

**Table 2.** Type and number of stores.

| Type | A | B | C | D | E | F | G | Sum |
|---|---|---|---|---|---|---|---|---|
| Number | 174 | 105 | 58 | 54 | 30 | 7 | 4 | 432 |

A: restaurant, pub, and party room. B: café, bakery, ice cream parlor, and rice cake shop. C: florist, clothing boutique, jewelry store, bookstore, perfume shop, vintage shop, and convenience store. D: office and atelier. E: beauty salon, barbershop, photo studio, skincare shop, nail salon, tattoo parlor, and tarot shop. F: gym, yoga studio, general practitioner, private academy, and real estate. G: cultural complexes and gallery.

### 3. Repurposed and Renovated Buildings and Changed Alleyways

*3.1. Activities in the Yeonnam-Dong Area and Repurposed Buildings*

Yeonnam-dong was a typical urban residential area. Most of the buildings were residential, and roadside buildings accommodated the neighborhood service facilities (laundry, supermarkets, and restaurants). There are now many restaurants and shops in the residential quarters as well. Commercial gentrification in this area is still underway; the area is getting wider, and the residential buildings are being repurposed and renovated for commercial use.

The nature of the trade zone in this area is reflected in the types of stores that have opened here. As of November 2020, according to the trade area analysis conducted by a company using credit card transactions, the proportions of food service and retail sectors account for 64% and 18.5%, respectively (Table 1), among the total number of shops. According to the frequency of KB card use by the floating population, the area is visited most frequently by women in their 20s (24.8%), followed by men in their 30s (12.6%) and 40s (7.2%), which indicates that the Yeonnam-dong area is popular among young people (and more among women than men) (Figure 4).

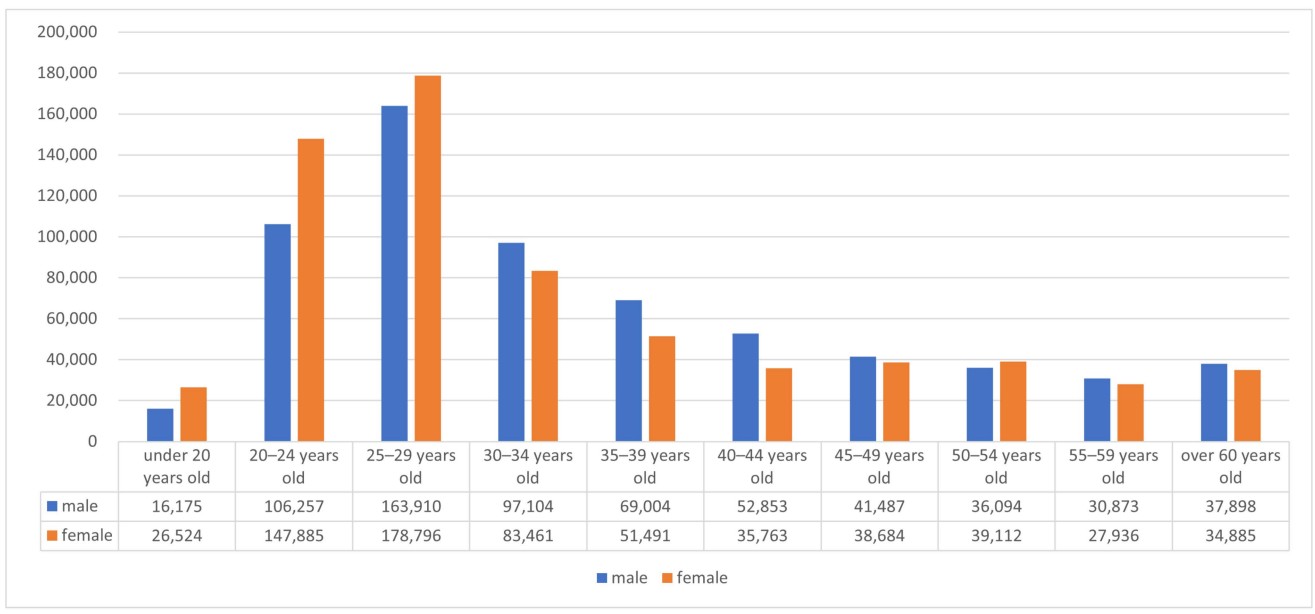

**Figure 4.** Customer distribution by age: F 20s (24.8%) > M 30s (12.6%) > M 40s (7.2%). F and M represent female and male populations, respectively.

The data collected in this study coincided with the types of the stores. The facilities accommodated by the 149 buildings surveyed include restaurants, pubs, party rooms, cafés, bakeries, florists, clothing boutiques, jewelry stores, various shops, bookstores, multicultural space, galleries, beauty salons, barbershops, photo studios, private academies, gyms, general physicians, yoga studios, offices, ateliers, and workshops. Leisure service, culture, and entertainment facilities dominate the area rather than neighborhood service facilities. The majority of facilities involve food and beverages. To identify the characteristics of the facilities and the patterns of cultural consumption activities of the floating population, we analyzed the shops in seven categories (types): food service facilities with liquor licenses, such as restaurants and pubs (type A), food service facilities with no liquor license, such as bakeries and cafés (type B), retail sector showing the details of the consumption made by the floating population (type C), offices and handicraft workshops (type D), service sector showing the details of the services received by the floating population (type E), fitness and health care (type F), and culture (type G). The food and beverage sector, consisting of shop types A (40.3%) and B (24.3%), account for the largest proportion (64.6%) of all

the facilities. The majority of restaurants and pubs serve a variety of foreign foods and refreshments rather than Korean food, indicating the local preference. Of the remaining facilities, the general merchandise stores (type C), such as florists, clothing boutiques, shoe stores, vintage stores, jewelry stores, and perfume shops, account for 13%; service facilities (type E), such as beauty salons, barbershops, photo studios, skincare shops, nail salons, tattoo parlors, tarot shops, account for 7%; offices (e.g., design, architect, interior design, entertainment, record producing, and logistics) and ateliers (e.g., jewelry, tailoring, and pottery) account for 12%.

Since there is little space to park in and around the area, young people who prefer public transportation come here. People who tend to avoid crowded shops downtown and seek small and unique shops also choose this area to shop and pass their time. They stroll through the side streets, purchase clothes, jewelry, and miscellaneous vintage goods, receive body care services such as nail art, tattoos, and hairdressing, and indulge in refreshments, such as coffee, exotic foods, and drinks, in restaurants, bistros, and bars. In addition to the shops and facilities, there are many small handicraft workshops (ateliers) for general merchandise supplies, along with offices related to culture and arts.

*3.2. Types of Building Renovation*

The repurposed buildings were renovated in a variety of styles (Table 3). There are no specific architectural styles or trends commonly found in renovated buildings. Each building reflects the commercial ideas of the building, and shop owners are keen to express their own personalities, which has resulted in a wealth of interior design and exterior styling arrangements. However, certain renovation types can be identified by comparing the original and renovated states because the original building type determines the scope of renovation. Therefore, the architectural characteristics and traits of a renovated building can be inferred from its prerenovation architectural type (Figure 5). Prerenovation buildings are grouped into the "home/shop combination", single-family housing, and multifamily housing. The home/shop combination type is a two-story, roadside building (with the shop downstairs and home upstairs). The single-family housing type is a detached, one-household building having a sufficient courtyard space, mostly built in 1984 or earlier. The multifamily housing type is typically a three or four-story building (with a residential basement finish permit without a courtyard), mostly built from 1985, with the relaxation of multifamily construction regulations. We divided each of these three architectural types into three subtypes according to the extent of renovation: (i) partial renovation with the original form retained, (ii) large-scale renovation with the original form partially retained, and (iii) large-scale renovation with the original form changed.

**Table 3.** Types of renovation of investigated buildings.

| Original Housing | Renovation Type | No | Sum |
|---|---|---|---|
| Home/shop combination | Partial renovation with original form retained | 15 | 15 |
| Single-family housing | Partial renovation with original form retained | 41 | |
| | Large-scale renovation with original form partially retained | 19 | 69 |
| | Large-scale renovation with original form changed | 9 | |
| Multifamily housing | Partial renovation with original form retained | 10 | |
| | Large-scale renovation with original form partially retained | 31 | 65 |
| | Large-scale renovation with original form changed | 24 | |
| | Sum | 149 | 149 |

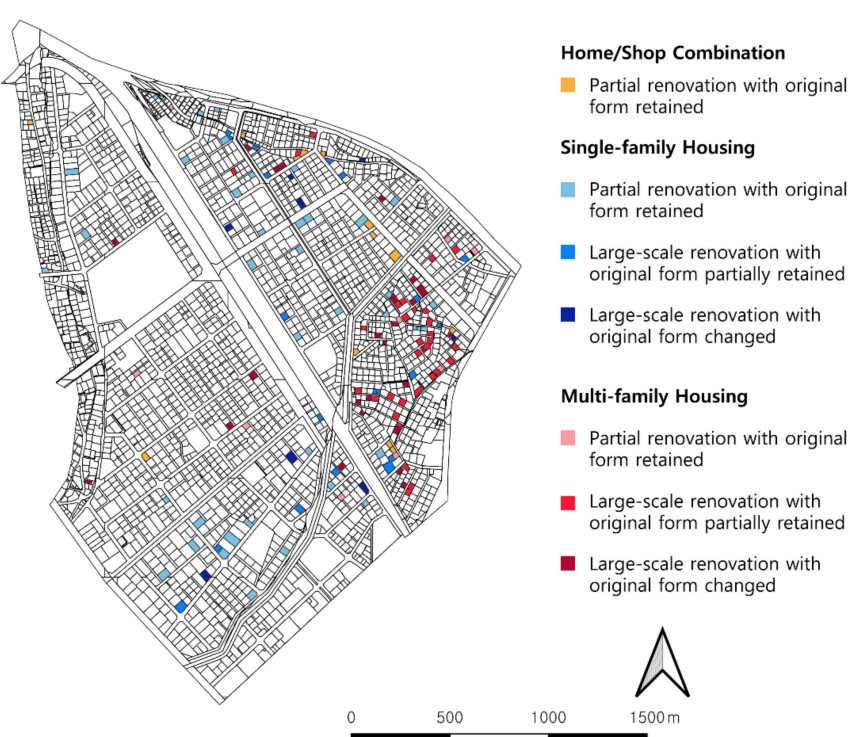

**Figure 5.** Distribution of buildings by renovation type.

The 149 buildings surveyed were divided into two architectural types: pre-1985, single-family houses constructed between the early 1970s and the mid-1980s, and post-1985, multifamily houses constructed between the mid-1980s and the late 1990s. A large portion of the multifamily buildings were found in the block having irregular alleyways between Dongjin Market and the subway station. This is the most commercialized area, with a large number of renovated multifamily housing buildings. The temporal distribution for the construction and renovation [year (n)] of this type of buildings is as follows: construction: 1966 (4), 1967 (4), 1968 (3), 1969 (6), 1970 (12), 1971 (4), 1972 (8), 1973 (9), 1974 (12), 1975 (3), 1974 (12), 1975 (3), 1976 (1), 1977 (2), 1978 (2), 1982 (1), 1983 (1), 1984 (2), 1985 (1), 1986 (2), 1987 (2), 1988 (2), 1989 (12), 1990 (14), 1991 (11), 1992 (10), 1993 (4), 1994 (3), 1996 (4), 1997 (1), 2001 (4), 2003 (1), 2005 (1), 2011 (1), unknown (2); renovation: 2004 (1), 2005 (3), 2007 (1), 2009 (1), 2011 (5), 2012 (3), 2013 (5), 2014 (11), 2015 (21), 2016 (28), 2017 (41), 2018 (25), 2019 (1), unknown (3).

We collected data for 15 shop/home combination buildings. They were all pre-1985 buildings that were planned to rent the first floor to community facilities, such as laundry, interior design studio, and grocery stores, providing living spaces on the second floor. After 2012, the community facilities began to be replaced with cafés, bars, restaurants, jewelry stores, and clothing boutiques, and renovations were made accordingly. The is distribution for the construction and renovation [year (n)] of these types of buildings are as follows: construction: 1967 (1), 1969 (1), 1970 (6), 1971 (1), 1973 (1), 1974 (1), 1982 (1), 1983 (1), 1984 (1), 1985 (1); renovation: 2012 (1), 2013 (1), 2014 (2), 2015 (2), 2016 (2), 2017 (4), 2018 (1), unknown (2).

The renovation of the aforementioned building 148-3 Donggyo-dong corresponds to this type. It was constructed in 1982 and its shop floor was occupied by the interior design studio of the community until 2009. It was thereafter rented to a small café. In 2014, this building was renovated into the current three-story building, of which the basement, first, and second floors were used for a café, and the extended third floor was a housing unit. As a whole, its original construction form was retained. The building was renovated without additional entrances, and the third floor was added. The side fence was demolished, and a new element was added with partial paint work and partial paneling on the façade.

This renovation created an ambiance that matched the vintage interior. The majority of combination buildings, with shop downstairs and residences upstairs, were renovated to maintain the overall original form (Figures 6 and 7).

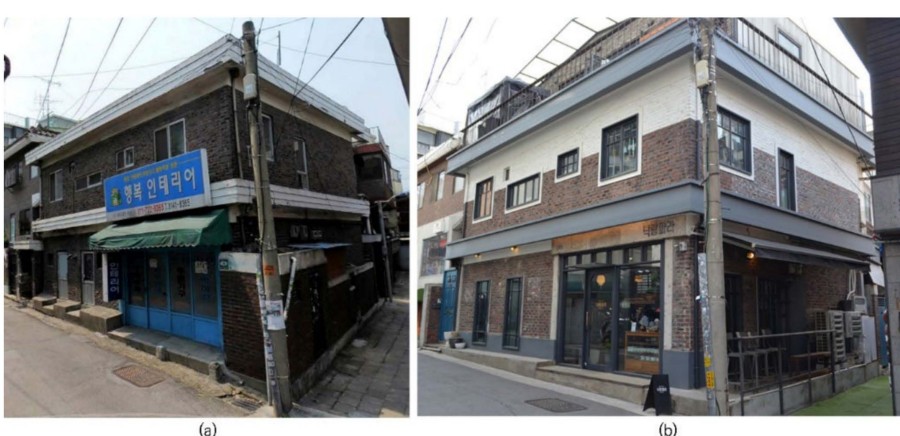

**Figure 6.** (**a**) Pre- and (**b**) post-renovation pictures of building at 148-3 Donggyo-dong.

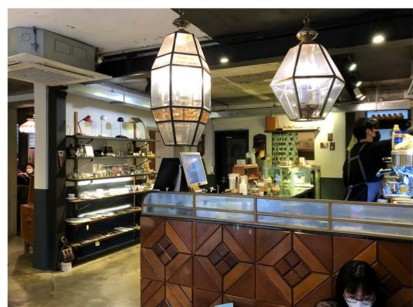 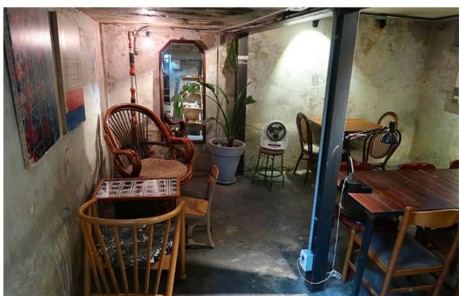 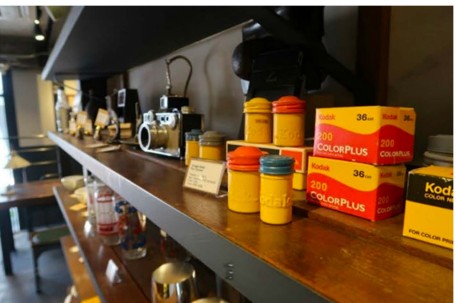

**Figure 7.** Renovated interior of building at 148-3 Donggyo-dong.

Of the 69 single-family houses mostly built in the 1960s and 1970s, 41 houses were partially renovated with the original form retained as the dominant renovation type. The temporal distribution for the construction and renovation [year (n)] of this type of buildings is as follows: construction: 1966 (4), 1967 (2), 1968 (1), 1969 (3), 1970 (2), 1971 (2), 1972 (3), 1973 (6), 1974 (8), 1975 (3), 1976 (1), 1977 (1), 1984 (1), 1986 (1), 1987 (1), 1992 (1), unknown (1); renovation: 2005 (3), 2007 (1), 2009 (1), 2011 (3), 2012 (2), 2013 (2), 2014 (5), 2015 (3), 2016 (5), 2017 (8), 2018 (6), 2019 (1), unknown (1).

Those built in the 1960s were smaller and deteriorated. Those built in the 1970s were typically three-story buildings, with the courtyard and first floor used for parking and the second and third floors being used as the living space. Of these 41 buildings partially renovated with the original form retained, 28 were renovated after 2014. The building at 254-8 Yeonnam-dong was a two-story building built in 1970. It had a courtyard and an entrance to a ground-level parking space. It was renovated to its current form in 2017. The ground-level parking space was converted into shops, and the second and third floors and rooftop were converted into a café and three small handicraft workshops having exhibition spaces (jewelry store, nail salon, and perfumery). The main renovation of this building involved the interior with the original form being retained. The entrances were enlarged, and stairs to reach above the first floor from the courtyard were installed. This had the effect of maximizing the indoor space for rent. Concurrently, the stairs and balustrades on the second floor and rooftop provide accents to the building. Such stairs and balustrades are typical renovations of a single-family house having a courtyard (Figures 8 and 9).

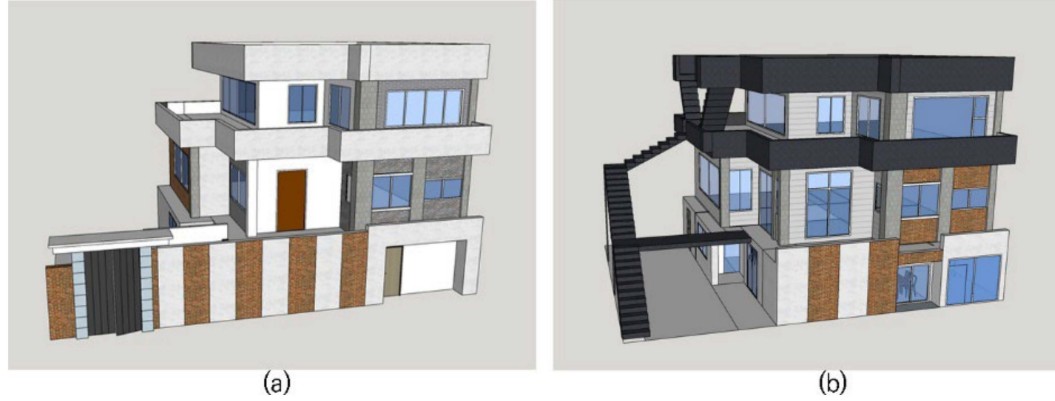

**Figure 8.** (**a**) Pre- and (**b**) post-renovation pictures of building at 254-8 Yeonnam-dong and small shops.

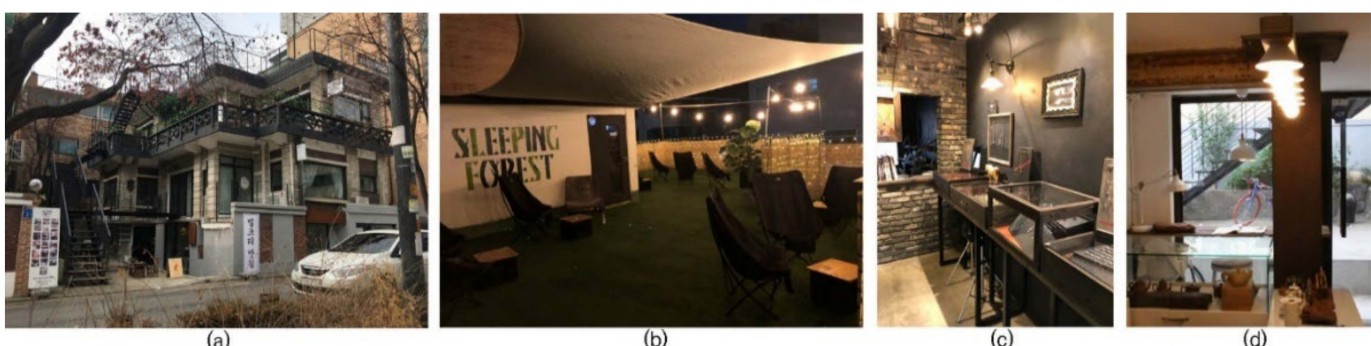

**Figure 9.** Building at 254-8 Yeonnam-dong and small shops: (**a**) complete view, (**b**) rooftop of café, (**c**) jewelry store on the second floor, and (**d**) café on the first floor.

Nineteen of the single-family houses were intensively renovated with the original form being partially retained. This renovation type typically involves adding a new architectural element or changing the colors. Most of them were constructed in the 1960s and 1970s and renovated after 2015. The temporal distribution for the construction/renovation [year (n)] of this type of buildings is as follows: construction: 1967 (1), 1968 (2), 1969 (2), 1970 (1), 1971 (1), 1972 (2), 1973 (2), 1974 (1), 1977 (1), 1978 (2), 1989 (2), 1990 (1), 2005 (1); Renovation: 2004 (1), 2011 (2), 2014 (1), 2015 (4), 2016 (5), 2017 (4), 2018 (2).

The building at 223-20 was constructed in 1972, and the entire building was renovated in 2018 into a bakery–café. The parking space was converted into the interior of the café accessible from the street, the gate was demolished, and the space was converted into a vestibule leading to the main hall of the café on the first floor. A fake wall was installed on the façade, thereby forming a special outdoor space on the second-floor terrace. The square fake wall of decorative bricks brought a new form to the building, while maintaining the original building façade, whose color was changed to a brighter tone. The interior paid little attention to decorative elements, but the rough brick finish (exposed by removing the ceiling) and the bright color finish attributed a unique charm to the place. The commercial charm of the café was staged by the diversity of interior and exterior spatial features. Compared with the small stores of the area, the café occupies ample indoor and outdoor spaces of the entire building (Figures 10 and 11).

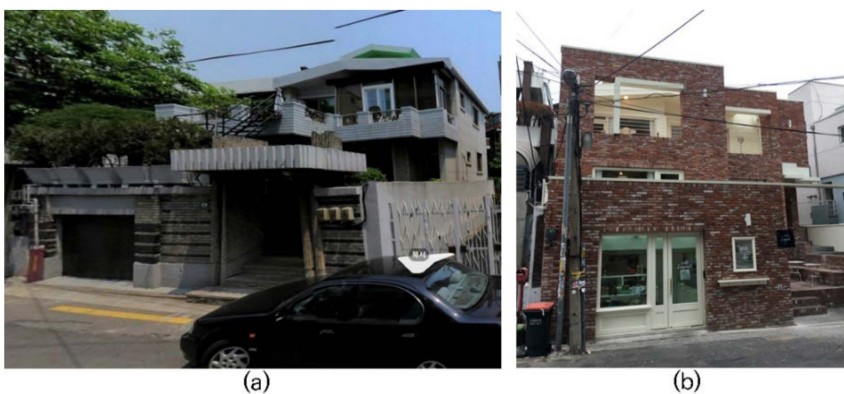

**Figure 10.** (**a**) Pre- and (**b**) post-renovation pictures of building at 223-20 Yeonnam-dong.

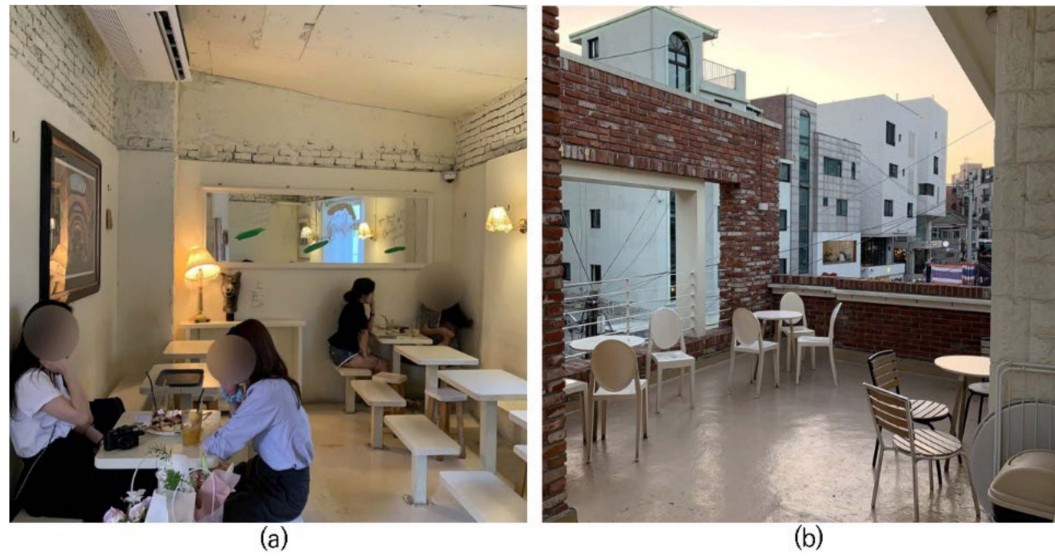

**Figure 11.** (**a**) Renovated interior and (**b**) terrace of the building at 223-20 Yeonnam-dong.

Nine of the single-family houses were intensively renovated, with the original form hardly retained. The temporal distribution for the construction and renovation [year (n)] of this type of buildings is as follows: construction: 1970 (2), 1972 (3), 1974 (2) 1990 (1), 2011 (1); renovation: 2013 (1), 2014 (1), 2016 (1), 2017 (3), 2018 (3).

Most of them were constructed in the 1970s. The building at 249-8 Yeonnam-dong (built in 1974) was a detached house with a typical courtyard, but the renovation in 2018 transformed it into a commercial building with a large-scale extension. Currently, the first floor accommodates an interior decor store and a clothing boutique; a Hanbok (Korean traditional cloth) rental store is on the second floor, and the third and fourth floors are used as living spaces. With the walls and gates demolished, the courtyard is open to the outside, and the exterior stairs provide access above the second floor, with separate entrances for each store and the housing unit. The original structure had no capacity for a core space, including stairs; the outside stairs contribute to maximizing the inside space. This is a case where the exterior of the house has been completely changed into a shopping mall through renovations, and the shops on the first and second floors show the nature of commercialization in this area. This building provides an example of a single-family house that is completely renovated into a commercial building (Figure 12).

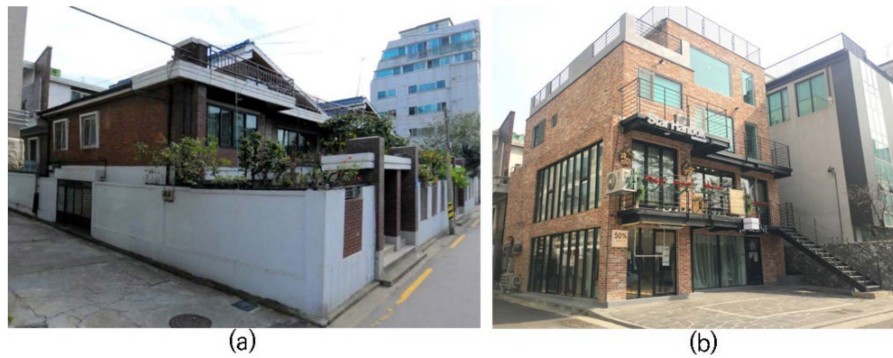

**Figure 12.** (**a**) Pre- and (**b**) post-renovation pictures of the building at 249-8 Yeonnam-dong.

Ten of the 65 multi-family buildings surveyed were partially renovated while retaining the original form. These building belonged to the post-1985 period and were renovated from 2015 onward. The temporal distribution for the construction and renovation [year (n)] of this type of buildings is as follows: construction: 1986 (1), 1989 (3), 1990 (1) 1991 (1), 1992 (2), 1993 (1), and 2001 (1); renovation: 2013 (1), 2015 (3), 2016 (3), 2017 (1), and 2018 (2).

The building at 387-24 Yeonnam-dong was constructed in 1992 and renovated in 2018, keeping the original shape intact. A multifamily house, it included a staircase to reach each floor, which was retained during renovation. A vintage look was staged by removing the balustrades of the stairs and exposing the unpainted concrete. Steel and glass were used to create the stairs, terraces, railings on the roof, and the roof of the stairs, giving an open feel to the building. New openings were added for each level. The main focus of renovation was exposing the interior and exterior spaces of the building. This building houses an Italian restaurant, chicken dish restaurant, wine bar, and cocktail bar. The interior of the wine bar on the second floor shows that the walls were removed, and the load of the slab was carried out using reinforced steel columns and beams. The steel columns and beams are fully exposed, as are the concrete floor of the slab and the cement mortar of the wall. The same interior style was applied to the Italian restaurant on the third floor and rooftop. Overall, a strong personal taste of the store owners is expressed with inexpensive interior finishes and accessories (Figures 13 and 14).

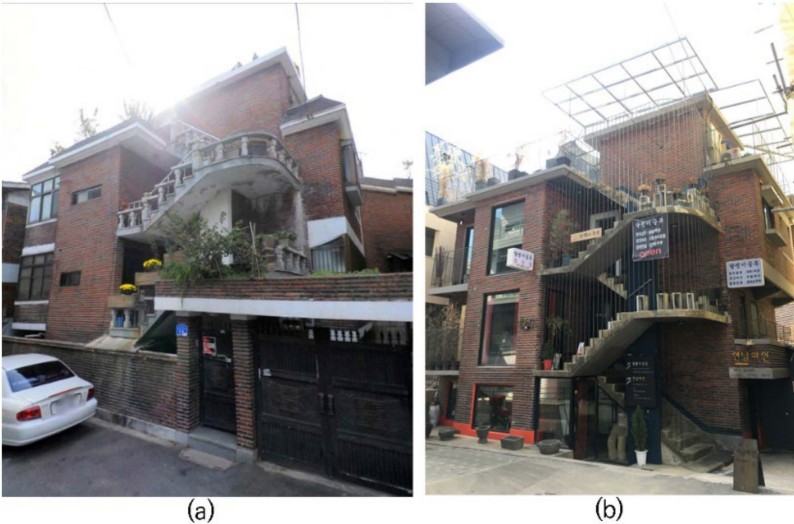

**Figure 13.** (**a**) Pre- and (**b**) post-renovation pictures of the building at 387-24 Yeonnam-dong.

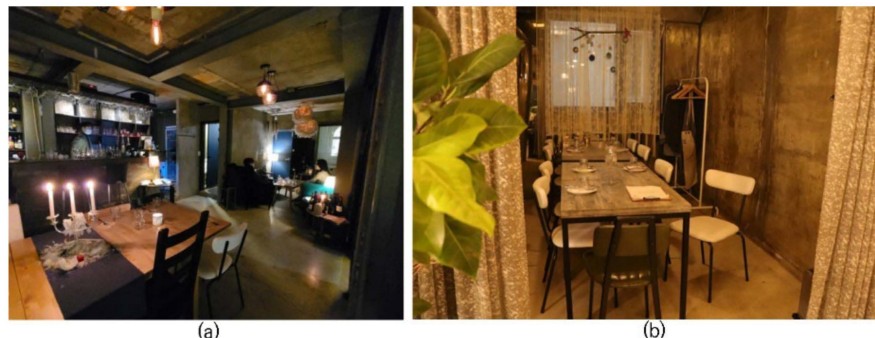

**Figure 14.** Renovated interior of the building at 387-24 Yeonnam-dong: (**a**) wine bar on the second floor, and (**b**) Italian restaurant on the third floor.

Thirty-one of the 65 multi-family buildings were extensively renovated while retaining the original form. This renovation type is most frequently observed in multifamily buildings. In this case, renovation involves substantial extensions or drastic changes, such as big openings and introducing new architectural elements. The temporal distribution for the construction and renovation [year (n)] of this type of buildings is as follows: construction: 1970 (1), 1988 (1), 1989 (5), 1990 (7), 1991 (8), 1992 (5), 1996 (1), 2001 (1), 2003 (1), and unknown (1); renovation: 2015 (7), 2016 (7), 2017 (11), and 2018 (6).

Most of these buildings were constructed in the late 1980s and early 1990s after the relaxation of multifamily construction regulations in 1985. With the permission of the basement's resident, a multifamily building typically uses the semi-basement as the ground floor and the two upper levels as living spaces. These multifamily buildings were renovated after 2015, with the kickoff of the full-scale commercial gentrification of the Yeonnam-dong area. The building at 147-48 Donggyo-dong was constructed in 1990 and renovated into its current state in 2015. The original structure had two entrances on both sides, with separate staircases. The outside staircases were retained, large front windows were added, and a black panel wall finish was incorporated, giving an image of a commercial building, while utilizing the brick structure of the original building. On the survey day, the building housed a Japanese restaurant and bakery on the first floor, a café and Italian restaurant on the second floor, and a sushi bar on the third floor. The areas of the shops sharing the first and second floors were very small (Figures 15 and 16).

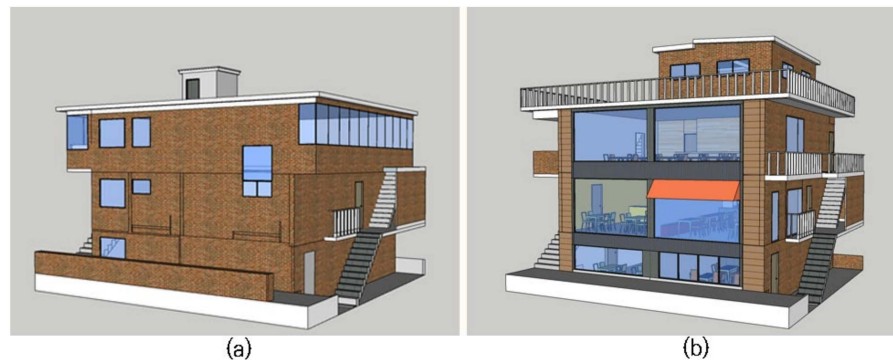

**Figure 15.** (**a**) Pre- and (**b**) post-renovation pictures of the building at 147-48 Donggyo-dong.

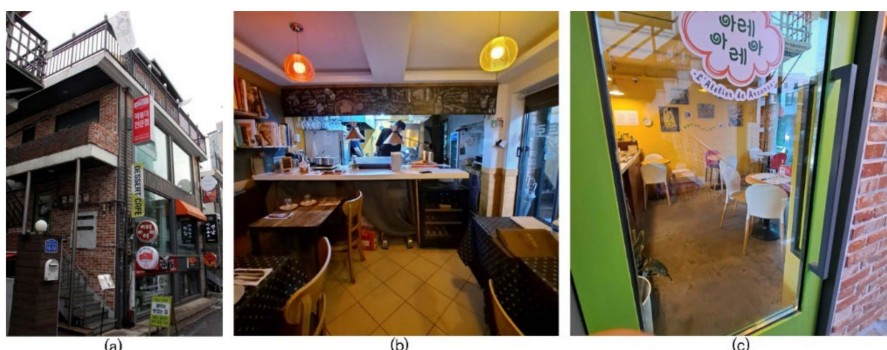

**Figure 16.** (**a**) Building at 147-48 Donggyo-dong and (**b**,**c**) the shops in the building.

Twenty-four of the 65 multifamily buildings were renovated substantially from the original form. Most of them were constructed in the late 1980s and early 1990s, and their renovation began in 2014, peaking in 2017. The temporal distribution for the construction and renovation [year (n)] of this type of buildings is as follows: construction: 1987 (1), 1988 (1), 1989 (2), 1990 (4), 1991 (2), 1992 (2), 1993 (3), 1994 (3), 1996 (3), 1997 (1), and 2001 (2); renovation: 2014 (2), 2015 (2), 2016 (5), 2017 (10), and 2018 (5).

The multifamily building at 147-26 Donggyo-dong was constructed in 2001 and renovated into its present form in 2016. The core space in the center of the building, which was used to provide access to each housing unit, was retained in the renovation design. Shops were installed on the first three floors, and the fourth floor was used as a housing unit. Large windows were installed on the façade, and the overall color scheme was changed. Thus, a multifamily building with interior stairs to each unit can be easily renovated to look like a commercial building by altering the façade design (Figure 17).

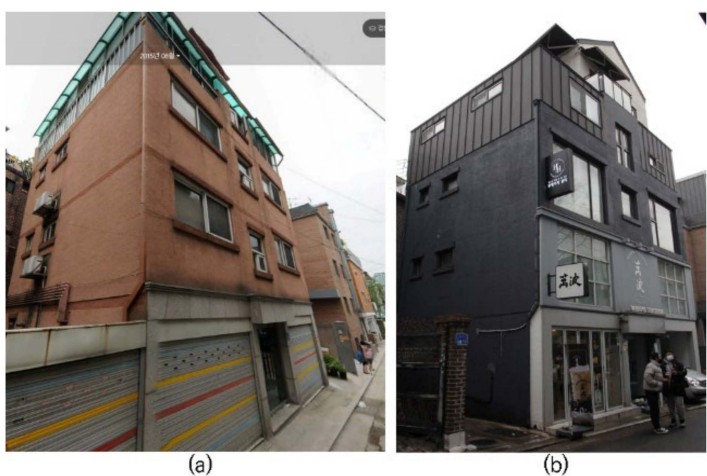

**Figure 17.** (**a**) Pre- and (**b**) post-renovation pictures of the building at 147-26 Donggyo-dong.

### 3.3. Transformed Alleyways as Public Spaces

The alleyway landscape of the Yeonnam-dong residential area consists of low-rise single- and multifamily houses with brick fences. With the advent of commercial gentrification, this landscape changed due to renovations. The fences were demolished, and rest areas (with tables and chairs) were established in the courtyards, along with decorative items. In addition, small empty spaces around the courtyard (or even rooftops) were converted into rest areas. Another important element that contributes to changes in the landscape is a renovated façade, with additional windows, diverse façade materials and colors, and signboards. In contrast to the pre-gentrification alleyway landscape (with low-rise red-brick houses and clear space divisions), the renovated landscape is unique

with harmonious residential and commercial aspects and co-existing private and public spaces.

Cases of courtyard space utilization after the removal of fences are often seen in pre-1985 single-family houses. Using a spacious courtyard as an outdoor rest area makes it appealing to passersby. The buildings at 227-34 Yeonnam-dong (constructed in 1968 and renovated in 2014) and 204-34 Donggyo-dong (constructed in 1974, renovated in 2009) house Italian restaurants, where the outdoor spaces are used for outdoor dining. The building at 454-11 Yeonnam-dong (constructed in 1966 and renovated in 2019) provides a space for people to linger, creating an open space between the alleyways, with simple landscaping in the courtyard. The renovation of the building at 245-67 Yeonnam-dong (constructed in 1984 and renovated in 2017) opened the entire courtyard and a steel staircase was installed, thereby offering a space for public use (Figure 18).

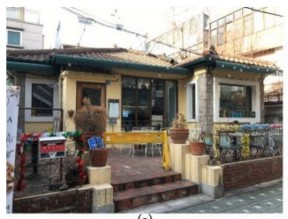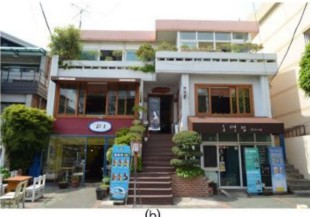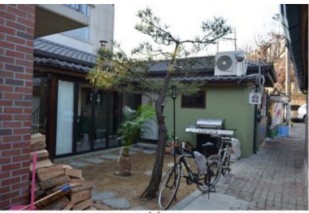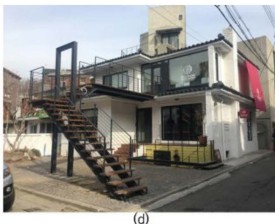

**Figure 18.** Examples of courtyard utilization: (**a**) 227-34 Yeonnam-dong (constructed in 1968, renovated in 2014), (**b**) 204-34 Donggyo-dong (1974, 2009), (**c**) 454-11 Yeonnam-dong (1966, 2019), and (**d**) 245-67 Yeonnam-dong (1984, 2017).

Even though they had no courtyards, the fences of several multifamily houses were removed, and the narrow spaces were utilized to showcase the owners' personal taste through decor items and plants. A commercial feel was incorporated in the look of the building by renovating the façade with new materials and colors, in contrast to the typical red-brick finish. The small dessert café in the building at 390-64 Yeonnam-dong presents a clean image, with a simple white finish surrounded by a black frame. The façade can be altered through window expansions, and new materials and design indicate the commercialization of the alleyways. In particular, expanded windows draw attention from the outside to ongoing commercial activities, thereby acting as an advertisement medium. The jewelry shop in the building at 148-2 Donggyo-dong has large windows on the street side, enhancing the presence of the shop; thus, even a small shop with an open design can change the ambiance of the alley (Figure 19).

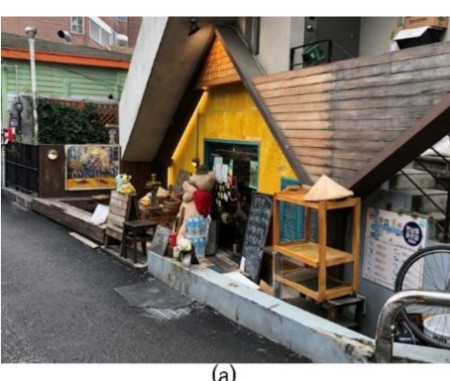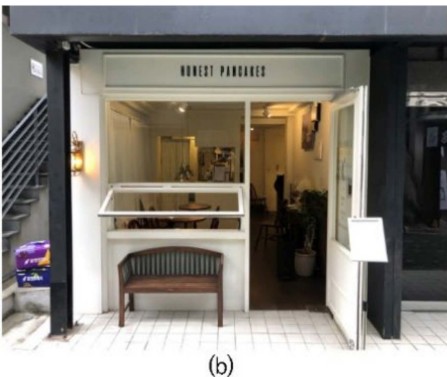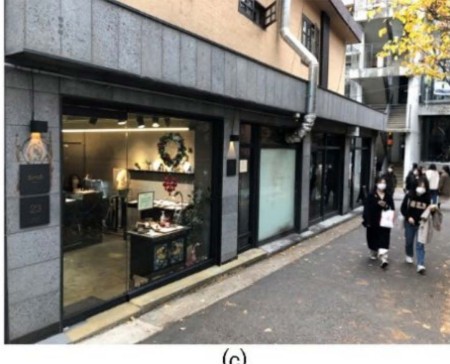

**Figure 19.** Examples of façade decoration ((**a**) 390-77, (**b**)390-64 Yeonnam-dong) and show window ((**c**) 148-2 Donggyo-dong).

A walk in this neighborhood has now become a cultural experience designed to enjoy the ambiance of a narrow and winding alleyway, while appreciating the various spaces,

open courtyards and decor items displayed by individual shops/businesses, window-shopping for clothes, jewelry, and miscellaneous goods, and peeking at the interior spaces of the cafés that line the alleys. This ambiance has become a signature of Yeonnam-dong.

## 4. Perception and Ambiance of Urban Architectural Changes

### 4.1. Visitors and Store Owners' Perception on Urban Architectural Changes

The interviews with visitors indicate their perceptions of Yeonnam-dong's urban and architectural changes. A total of 31 visitors were asked about Yeonnam-dong's features and their reasons for visiting, in response to which many spoke about Yeonnam-dong's unique ambiance. Thirteen of the 31 said that Yeonnam-dong has a unique ambiance, using the words "ambiance", "feeling", and "sensibility". Fourteen of 31 visitors cited diversity as a characteristic of the area: "there are many unique shops"; "there are many good shops and it is tranquil, so it is good for walking"; "there are many beautiful places and restaurants serving delicious food, so it is good for taking pictures and meeting friends"; "there is much to see while walking along the streets"; "there are many new things to see everywhere"; "it is fun to look around the various individually owned shops and walk through the alleys". Here, the diversity of shops reflects the area's locality. In addition to the diversity of shops, the diversity of exterior decorations and interior spaces is also revealed. Various architectural elements have been incorporated to reflect personal taste or a commercial look. Here, urban characteristics also contribute to the overall ambiance. Five out of 31 visitors used the expressions "stroll", "walk", and "look around". Eight out of 31 visitors voiced enjoyment of or criticized the residential environment: "the alleys themselves are cute and dainty"; "the buildings are not tall"; "it is quiet and small"; "it is narrow because of the residential area". A man in his 20s stated that he visits this place two to three times a month and often dines at Japanese restaurants. His perception of the area clearly reflects Yeonnam-dong's locality: "Since it is next to a residential area, it has a quieter and cozier ambiance than places like Hongdae. One advantage is that it has few franchise shops and many unique stores. I like the cute and natural feel, but I do not want it to be excessive."

Thus, although Yeonnam-dong is a residential area that is less commercialized than the bustling commercial area at Hongdae, there are some shops with an excessive commercial feel. Another man, who is in his 30s and works in the area, also expressed opinions about excessive commercial ambiance: "My friends also live in Yeonnam-dong. The alleys are complex yet simple, making it fun to walk. The peculiar mix of people staying in the area and those coming and going makes you think about many things. I do not think much about the renovated buildings, but I get the impression that the new buildings being built recently are uniform."

As someone who has worked and lived here, he is aware of changes in the area and mentioned the staleness of commercial expression of newly constructed buildings. Thus, visitors perceive Yeonnam-dong's unique ambiance with narrow alleys in uncrowded low-rise residential areas, through its various small shops and rapidly progressing commercialization.

The store owners hold the same perceptions of the area. The owner of the café at 148-3 Donggyo-dong had renovated the two-story building to three stories and opened a café in 2014. Thus, he is one of the gentrifiers from 2014 to 2015, when the block around Dongjin Market was rapidly commercialized. A former practitioner in the interior design industry, he possessed sufficient capital to buy and renovate an entire building. We classified this building as a renovated shop/home combination building. He is aware of the area's characteristics of an old-fashioned ambiance with many alleyways, and decorated the café to match the vintage ambiance: "I moved from Daechi-dong to Yeonnam-dong in search of somewhere that felt like it had history rather than somewhere modern and where I could run both an office and café. Daechi-dong is located in Gangnam, Seoul. Gangnam's development began in the 1980s, and it now has the highest concentration of capital and wealth in Korea. There are many alleys, and the old buildings contribute to Yeonnam-

dong's ambiance. I liked the artistic ambiance and commercial area, and, although the old-fashioned simple ambiance is gone, it is still quite fine. I tried not to touch the aged brick houses as much as possible. I used vintage props and antique furniture that I had collected."

As another example, the shop owner of the small dessert café selling French confectionery in 254-8 Yeonnam-dong, southeast of Gyeongseong High School and close to Gyeongui Line Forest Park, has a similar but slightly different perception of the area. It is because, unlike the Dongjin Market area, which has narrow and irregular alleyways, the building is located in a residential area on a wide road and well-kept lot and is close to Gyeongui Line Forest Park. We classified this building as a single-family house partially renovated with the original form retained. While generally retaining the ambiance of the residential area and the appearance of an old building, he decorated the interior with an understated design to match the renovated building: "As this is close to the Hongdae commercial district and a quiet and well-kept location next to Gyeongui Line Forest Park, I decided to open a store here. I tried to create a classic aura when remodeling the interior."

Conversely, shop owners who recently entered the buildings inside the Dongjin Market block in 2020 have a slightly more commercial perspective. These are Western restaurant owners who moved into 383-23 Yeonnam-dong and 383-72 Yeonnam-dong. We classified these buildings as multifamily houses extensively renovated with the original form partially retained. They are located on Yeonhui-ro 1-gil, which is regarded as the most commercialized street, where entire buildings have undergone glamorous renovations. These two shop owners shared the following: "It is an attractive location. It has infrastructure for restaurant businesses and a large floating population thanks to Hongdae, Gyeongui Line Forest Park, and the various commercial areas. Despite the intense competition as new stores open and close very quickly, it is still an attractive commercial area. There are many cases of old buildings being renovated and changed to different industries and then the area converting to a shopping district. Recently, there are many buildings being demolished and newly built, so the feeling of the alleyways is becoming ambiguous. We pursued a harmonious design that did not clash with the building overall." "I moved here because it was the most popular place in Seoul for the store price. It used to be a place of diverse culture where every spot had its own unique individuality, but that individuality and originality are now changing toward popularization. We pursued an original interior design for the building and tried to make photos stand out on Instagram."

Although they are aware of the ambiance of the alleyways and diverse commercial cultures in the area, the shop owners are thinking in terms of commerciality. For the interior design as well, they focused more on commercial effect rather than expressing their individuality. They also believe that the ambiance of the area is changing as it becomes increasingly commercialized.

The interviews with visitors and store owners revealed that the owners are taking advantage of the low-rise residential area in Yeonnam-dong and creating an ambiance with unique building renovations and diverse shops; thus, as the area is gradually being commercialized, its current diversity and individuality are being further weakened (Figure 20).

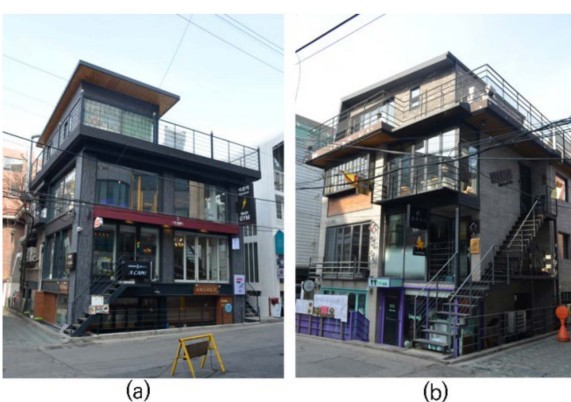

**Figure 20.** (**a**) Buildings of 383-23 Yeonnam-dong; (**b**) buildings of 383-72 Yeonnam-dong.

### 4.2. Ambiance of Urban Architectural Changes

We determined that Yeonnam-dong has a certain ambiance and placeness, which are changing due to commercialization of the area. This represents a transformation in the place's identity. The ambiance of Yeonnam-dong has been formed by accumulating activities, artifacts, and their implications through time.

First, on examining the history of the urban environment, it can be understood that early gentrifiers selected Yeonnam-dong because it was close to the Hongdae commercial area and the rent was inexpensive. However, Yeonnam-dong was also underdeveloped and had inconvenient commuting means. Its crooked and narrow alleyways, streets, and lots, subjected to land readjustment in the 1960s and 70s, and single-family, multihousehold, and multifamily houses of the 1960–1990 period symbolize underdevelopment and poverty. Thus, rather than mainstream apartments and supermarkets, the region was characterized by underdeveloped and small-scale infrastructure [12]. Nevertheless, the urban and architectural conditions of this still undeveloped place call to mind the past, and, as Fouser described, alleyways contain memories, induce nostalgia, and allude to the romantic, mystic, and exotic qualities of the place [25]. Thus, Yeonnam-dong offers a physical environment allowing gentrifiers and visitors to experience the history of the place while strolling through the alleyways of low-rise residential areas.

In addition to these urban conditions, gentrifiers established a new ambiance with interior and exterior decorations and various commercial activities. Starting in 2008, their migration to Yeonnam-dong caused commercial gentrification to spread from the south to the southeast of Gyeongseong High School. This area is unlike the irregular alleys of the Dongjin Market area; a land division from the 1960s and 1970s, it has an urban structure with relatively large lots and roads in a grid layout. This area continues to feature many single-family houses with yards on large lots; in these buildings, the survey revealed many renovations where the original building form was largely retained. In the entire survey area, of the 69 single-family houses, there were 41 such partially renovated buildings with original the form retained. The building at 197-40 Donggyo-dong (constructed in 1972, renovated in 2011) is a prime example. It is a two-story single-family house with a relatively large yard and a charm reminiscent of the past that is being used commercially. The early gentrifiers included cultural and artistic practitioners, who rented out parts of the building and renovated it to create a relevant ambiance to their enterprise, which contributed to the main ambiance of the area. Economically, this renovation type is associated with their low investment capacity. It is a "natural and not excessive" renovation, as described by one visitor. The building at 254-8 Yeonnam-dong, renovated in 2017 as described earlier, is a continuation of this case. Although its original form is a home/shop combination rather than a single-family house, the intention behind the renovation of 148-3 Donggyo-dong is also an illustrative example. The owner of this café tried to leave the old brick finish intact and designed the interior in a vintage style. Despite having sufficient capital, the owner intentionally sought to create a vintage ambience throughout the building. This is a

case of utilizing the area's past ambiance for commercial effect. This consumption of the past is referred to as "retro consumption". It is the consumption of heritage that enables one to experience history, difficult-to-attain scarcity, and a charm that distinguishes it from mass-produced mainstream products [26]. The spatial practices of small business owners in Yeonnam-dong arise from their preference for nostalgia, authenticity, and naturalness, which influence the creation of place. This cultural practice and consumption can be interpreted as a trend by shop owners to establish their own character that is distinct from the social mainstream; it is interpreted as a behavior to distinguish themselves. Yoon H. S. viewed the young small business owners of Yeonnam-dong as members of the new middle class, a group with limited economic capital and abundant cultural capital within the urban middle class, and then interpreted their tastes and spatial practices from the perspective of Bourdieu's distinction [27].

In this context, even in the renovated buildings of Yeonnam-dong, the main trend during initial gentrification was to renovate without excessive changes to retain the building's original form as much as possible. This formed the gentrification landscape, and, along with Yeonnam-dong's urban environment and the early gentrifiers' various activities, the initial "aura of authenticity" was established.

The café owner at 148-3 Donggyo-dong said that the Yeonnam-dong area of today does not have a small old-fashioned ambiance. The statements of the store owner at 383-23 Yeonnam-dong (new shops are opening and closing very rapidly), the store owner at 383-72 Yeonnam-dong (individuality and diversity are becoming popularized), and one visitor (the newly renovated buildings are uniform) indicate that the area is being rapidly commercialized, and its diversity and individuality are beginning to vanish. In particular, the blocks with irregular alleys between Dongjin Market and Gyeongui Line Forest Park are the most commercialized. The lots of these blocks are places where numerous multihousehold houses are distributed in relatively small lots. Many of the surveyed multihousehold residential buildings are distributed in this area. Of the 65 renovated buildings that were originally multifamily houses, 31 underwent large-scale renovation with the original form partially retained, and 24 underwent large-scale renovation with the original form changed. This shows that considerably active building renovations took place. Those that were originally multi-household buildings were two to three stories tall and on small lots for economic efficiency with barely any yards; thus, they lacked the charm of single-family houses with yards on large lots. Therefore, there were many cases of aggressive renovation involving addition of wider windows occupying a large portion of the building façade, prominently decorated stair rails and panels on the façade, or incorporation of various colors in the building, to enhance commercial appeal. In particular, for the buildings that underwent large-scale renovation with the original form changed, it is difficult to recognize their original form due to the addition of various forms and construction of extensions. Along with single-family houses that underwent large-scale renovation with the original form changed, this renovation type consists of residential buildings aggressively transformed into commercial buildings. Unlike typical commercial buildings in downtown centers, in a neighborhood with an ambiance reminiscent of the past with undeveloped low-rise residential buildings and alleyways, these types of renovated buildings are perceived by visitors and store owners as excessive, uniform, and popularized. This reflects the increasing instances of commercialization in the area, which are transforming the ambiance and place causing it to lose its initial "aura of authenticity".

## 5. Conclusions

Commercial gentrification has been underway in Yeonnam-dong since 2008, when cultural and artistic practitioners active near Hongdae could no longer afford the rising rent and moved to the areas south and southeast of Gyeongseong High School. Hongik University Station's opening on the Airport Railroad Express (2010), its phased opening on the Gyeongui–Jungang Line (2012–2015), and the completion of Gyeongui Line Forest Park (2015) led to rapid commercialization. Commercial gentrification began under the

influence of the Hongdae commercial district, a cultural consumption area, and was accelerated due to increased transportation accessibility and the park's completion. In other words, Yeonnam-dong's commercial gentrification corresponds to culture, traffic, and green gentrification, which combine cultural factors such as being located close to the Hongdae commercial area, and factors relating to urban infrastructure such as transportation and park installation. Additionally, cultural trends such as nostalgia occurred in the 2010s, and the urban physical characteristics of Yeonnam-dong, which lagged in the development, with alleyways and low-rise houses that stimulate this retro atmosphere, act as a factor in its commercial gentrification.

During early gentrification, cafés, restaurants, and pubs coexisted with offices, studios, workshops, and the like; even in 2014, the residential ratio was high and cafés and restaurants were increasing in neighborhood-type shops. Nevertheless, the area was showing signs of the initial stage of gentrification, where there are almost no chain stores. However, in 2018–2019, when the survey was performed, restaurants, cafés, and pubs comprised over half of the total shops, and businesses such as jewelry stores, perfume shops, and clothing boutiques had increased. Thus, Yeonnam-dong has been largely commercialized into a cultural consumption area with mainly exotic restaurants; however, its commercial diversity is being maintained as there are still barely any chain stores.

Within the commercialization process, small business owners' spatial practices transformed the area's streets and buildings, creating a new ambiance and place. They demolished stone fences, opened yards, and decorated the front of stores with chairs, tables, and props, thereby increasing the appeal of outdoor spaces. The business owners introduced a commercial feel by decorating the façades and renovating the interiors of existing single-family, multifamily, and multihousehold houses. Many of the surveyed single-family houses underwent renovations that retained the building's original form; these renovations created a non-exorbitant, natural, and authentic ambiance reminiscent of the past with a small investment. Visitors and store owners perceive Yeonnam-dong as a low-rise residential area with alleyways and quiet, cozy, small, and varied shops. The trend during early gentrification was building renovation and interior design that evoked retro or vintage sensibilities to match the area's image, thereby giving authenticity to the local ambiance. Thus, it means that, in a nostalgic urban environment, the unique gentrification aesthetics, sincerity, and placeness of Yeonnam-dong developed with the tastes and practices of the gentrifiers. However, after rapid commercialization took place, strong commercial expression appeared in buildings, primarily in renovated multihousehold residences. Many of the surveyed buildings had required larger investments for renovation than that required o renovate single-family houses and barely retained the original form. The goal was to create a commercial look through aggressive renovation and redesigned interiors; however, some visitors found it to be excessive, uniform, and popularized. This indicates the transition from the area's early unique, diverse, and authentic ambiance to an excessively commercial ambiance. Thus, as commercialization progressed, the initial gentrifier was replaced by the succeeding gentrifier, and, as the size of the capital invested in the region increased, the scale of building renovation became larger and more colorful, making the alleyways and local scenery more commercial. Additionally, visitors and store owners are aware of these changes and are questioning the aesthetics of local placeness and authenticity.

The process of commercial gentrification In Yeonnam-dong and the renovated buildings reflect political, economic, social, cultural, urban, and architectural undertones. The main agents of change in Yeonnam-dong are not the state and large corporations that have created large-scale apartment complexes, neither are they ordinary small business owners or businessmen with capital in the affluent Gangnam area of Seoul. These agents comprise the new middle class, the creative class who opened stores with small capital even though they had a significant amount of cultural and social capital. Therefore, their tastes and practices are different from those of the mainstream Korean society as reflected in the renovated buildings in Yeonnam-dong. These buildings are different from standard apartments and ordinary commercial buildings in existing commercial areas. New spatial

uses and decorative devices added to residential buildings built with the urban rules and architectural forms of the 1970s to 1990s are unique and meaningful. A combination of past heritage and modern needs, the Korean situation and exotic culture, and the façade of residential and commercial buildings has become a pronounced culture of the times. These physical features include the social status of the subjects, the past and present daily life, and the possibility of the future. These create "distinctiveness", creating a unique locality and authenticity of a region, and that authenticity is currently "selling".

As commercialization progressed in Yeonnam-dong, its difference and authenticity faded. This means that numerous gentrifiers have been replaced, and the aspect of renovation has changed. Thus, visitors recognize that the area is generalized like other areas. Active intervention by local governments is needed to halt the disappearance of the locality. It cannot entirely follow market principles alone. Local governments should serve as anti-gentrifiers. Governance should be established with building owners and tenants, local governments and experts, and institutional and administrative regulations, and support should be implemented. Although Seongsu-dong in Korea is an example of active intervention by local governments, achievements are partial and limited. Thus, continuous monitoring and analysis of its progress and results are needed [27].

**Author Contributions:** Conceptualization, J.-Y.L.; methodology, J.-Y.L.; software, D.W.A.; validation, D.W.A.; formal analysis, J.-Y.L.; investigation, J.-Y.L.; resources, D.W.A. and J.-Y.L.; data curation, D.W.A. and J.-Y.L.; writing—original draft preparation, D.W.A. and J.-Y.L.; writing—review and editing, D.W.A. and J.-Y.L.; visualization, D.W.A. and J.-Y.L.; supervision, J.-Y.L.; project administration, D.W.A. and J.-Y.L.; funding acquisition, J.-Y.L. All authors have read and agreed to the published version of the manuscript.

**Funding:** This work was supported by the Ministry of Education of the Republic of Korea and the National Research Foundation of Korea (NRF-2018S1A5A8030315).

**Institutional Review Board Statement:** Ethical review and approval were waived for this study due to submission of the documents, signed by the interviewers.

**Informed Consent Statement:** Not applicable.

**Data Availability Statement:** Not applicable.

**Acknowledgments:** The authors would like to express our thanks to the students Seok-hyeon Ki, Yeong-chan Kim, Joon-soo Park, Shin-beom Lee, Hyeon-joo Shin, and Jeong-jae Lim, who performed the field surveys for this study.

**Conflicts of Interest:** The authors declare no conflict of interest.

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
