# Peer review of "Implications of Renovated Buildings in Yeonnam-Dong, Seoul, an Area under Commercial Gentrification"

_sustainability, doi:10.3390/su15031960_

Round 1

Reviewer 1 Report

This is a good paper for  renovated buildings in commercial gentrification.just some small questions as following:

1.      The main question addressed by the research.

(1)what are the implications of renovated buildings in this paper?The author may be able to better define the "so-called impact" and environmental scope, and through scientific research methods, identify the factors that cause this impact?

(2)In the literature analysis, what are the differences and reference significance of similar experiences in other countries or regions?

(3)At the end of the paper,what are the good suggestions for losting distinctiveness?

2. The topic original in the field is general, and similar studies are common.

3. Compared with other published material, the paper aimed to identify the characteristics of the changes in the buildings and alleyways in Yeonnam-dong, Seoul, which has certain local characteristics about commercial gentrification pace.

4. This paper adopts a certain survey method to conduct a detailed investigation and study of the region, However, the conclusion needs further in-depth analysis. The authors should give better suggestions or countermeasures to the commercial phenomena in the region.

5. Are the conclusions consistent with the evidence and arguments presented and do they address the main question posed?

6. References can further supplement the experience of other regions

7. The tables and figures in it are fine.

Author Response

Reviewer 1

  1. (1) The meanings that renovated buildings contain are newly explicated in abstract and

 conclusions. Additionally, urban and socio-cultural factors on the gentrification of this

 area are complemented in conclusions.  About the future impact, I am not able to

 mention.

(2) In the literature analysis, the differences on gentrification between the occidental and the Korean are added. For the common phenomenon, the case of Montreal is explicated on the subject of authenticity.

(3) In conclusions, the evolvement of local government for the are suggested. But I can’t mention concrete policies in thinking of the quantity for conclusions. 

  1. In the conclusion, I suggest that it is necessary for policies of anti-gentrification by the local- government.
  2. For the main points that this paper insists on, the conclusions are revised.
  3. The papers on the cases of Montreal in Canada, Sungsoo-dong in Korea are supplemented in the references.
  4. Figure 1, 2, 3, 4, 5, and Table 4 and are improved for the good view.

Thank you very much for your review and comments. If you reindicate the things that should be improved, I will revise them.

Reviewer 2 Report

This paper studies the significance of the renovation of buildings in the commercial aristocratic area of Yeonnam-dong, Seoul, the main problems in this paper are as follows:

1. What are the innovative points of this paper, which should be clearly presented in the abstract and introduction.

2. The clarity of the pictures in this article needs to be improved.

3. The references cited in this paper are old, it is suggested to quote more references in the past five years.

4. The format of this article still has some problems and needs to be modified.

5. Overall, the author of this paper has done a good job and recommends that it is accepted after minor revision.

Author Response

Reviewer 2

  1. For the innovative and main points that this paper insists on, abstract and chapter 1.2 in introduction, and conclusions are revised.
  2. The pictures that show the past (before the renovation) can’t not be replaced by the improved. Because they are downloaded from the road view of some sites. The solutions of these pictures are limited.
  3. Several papers, published within 5 years, are added in the references.
  4. I don’t exactly know “the format” that you mentioned. The general contents are maintained. Could I ask for your understanding?

Thank you very much for your review and comments. If you reindicate the things that should be improved, I will revise them.

Reviewer 3 Report

The article is written correctly. I think adding a example to the gentrification definition in introduction would be valuable. Some picture, graphic. Gentrification is a fairly new term and not everyone stil feels it properly. The only complaint I have is too few references to their places where this phenomenon is observed. A summary would be a good place. In my opinion, this is a very important article.

Author Response

Reviewer 3

I revised the abstract and the chapter 1.2 in the introduction, and the conclusion. For the explanation on the gentrification, I complement the theoretical review in the chapter 1.2.

Thank you very much for your review and comments. If you reindicate the things that should be improved, I will revise them.
